# A New Statistical Method for Determining the Clutter Covariance Matrix in Spatial–Temporal Adaptive Processing of a Radar Signal

**DOI:** 10.3390/s23094280

**Published:** 2023-04-26

**Authors:** Adam Kawalec, Anna Ślesicka, Błażej Ślesicki

**Affiliations:** 1Faculty of Mechatronics, Armament and Aerospace, Department of Anti-Aircraft Missile Sets, Military University of Technology, 00-908 Warsaw, Poland; adam.kawalec@wat.edu.pl; 2Institute of Navigation, Polish Air Force University, 08-521 Dęblin, Poland; a.slesicka@law.mil.pl; 3Department of Avionics and Control Systems, Faculty of Aviation Division, Polish Air Force University, 08-521 Dęblin, Poland

**Keywords:** space–time adaptive processing, sample matrix inversion, radar MIMO

## Abstract

In this article, a new statistical method for estimating the clutter covariance matrix in space–time adaptive radar signal processing (STAP) is presented and studied. The new method was designed for multiple-input–multiple-output (MIMO) radar with time division multiplexing (TDM). An extensive analysis of statistical and non-statistical methods for estimating the clutter covariance matrix in STAP is presented in this paper. In addition, the STAP algorithm for the standard statistical SMI clutter covariance matrix estimation method, which is based on QR distribution, has been presented. The new method is based on LU distribution with partial pivoting. Simulation results confirm the validity of the presented model and theoretical assumptions. In addition, more accurate object detection results were demonstrated for specific computational examples than for other statistical methods. Considering the current analysis of the literature, it is noted that attention has now been focused worldwide on the study of non-statistical methods for estimating clutter covariance matrices in heterogeneous environments. Hence, it should be emphasized that the posted study fills a gap in current research on STAP.

## 1. Introduction

Nowadays, radar is finding more and more applications. In general, radar is a device for acquiring information about physical objects and their surroundings based on signals received from sensors that are not in direct contact with these targets. Radar is no longer just a device for detecting targets and the distance to them using electromagnetic waves. It is also a device capable of measuring the speed and direction of a target, carrying out identification, or creating an image of the earth’s surface. For this reason, the development of new algorithms related to radar signal processing technology is required [1,2,3].

The space–time adaptive processing (STAP) technique is used to detect targets moving on the earth’s surface through a radar system mounted on a flying platform. This technique allows the detection of moving targets against the background of two-dimensional interferences. STAP refers to the augmentation of adaptive antenna techniques to processors that simultaneously combine the signals received on multiple elements of an antenna array (the spatial domain) and from multiple pulse repetition periods (the temporal domain) of a coherent processing interval. This technique allows the detection of moving targets against the background of two-dimensional interferences [4].

The key step in the STAP algorithm is the correct determination of the clutter covariance matrix. In the literature on the subject, it can find various methods of estimating the clutter covariance matrix. The general division into statistical and non-statistical methods is applied [5].

The most popular statistical method is the SMI method. This method is most often the reference method for newly proposed methods. In statistical methods, the clutter covariance matrix is determined using data from training cells surrounding the cell under test [6].

Non-statistical methods mainly operate on a single cross-section of the radar data cube, thus reducing the computational complexity. In addition, they are effective for both homogeneous and heterogeneous clutter [7]. On the other hand, the use of non-statistical methods sometimes results in a decrease in performance relative to statistical methods.

MIMO radar refers to a radar system with multiple transmitters and multiple receivers. Compared to traditional phased array radar, MIMO radar can transmit multiple independent measurement waveforms at the same time, making it possible to increase the diversity of waveforms.

According to the array configurations, two types of MIMO radar should be distinguished: statistical MIMO radar and coherent MIMO radar [8,9,10]. Statistical MIMO radar refers to a system with widely separated transmit and receive antennas. The spatial diversity provided by statistical MIMO radar can be used to achieve higher localization accuracy and to cope with target amplitude fluctuations [11,12]. Unlike statistical MIMO radar, the antennas of coherent MIMO radar are collocated; hence, it is known as collocated MIMO radar. Compared with phased array radar, coherent MIMO radar can achieve a higher number of degrees of freedom (DOF) [13,14].

Thanks to its excellent target detection performance and parameter estimation, MIMO radar has recently found many applications, which include autonomous driving [15], remote sensing [16], landmine detection [17], and disaster management [18].

## 2. Related Works

Among the statistical methods can be distinguished methods involving the deliberate selection of training bins [19,20,21]. In article [20], it was described as a method of training sample selection (training sample selection), which consists in accepting only those training cells for which the covariance matrices are similar in a certain way to the covariance matrix of the bin being tested for the presence of a target.

Another method used to select training bins to determine the clutter covariance matrix is the Generalized Inner Product (GIP) method. STAP based on the GIP method may inaccurately estimate the clutter covariance matrix, leading to degradation of STAP performance [20,22].

Most of the known basic statistical methods associated with the selection of appropriate training bins described above have modifications aimed at eliminating undoubted disadvantages that negatively affect the performance of a given STAP method. Over time, statistical methods based on the above have been developed, such as Prolate Spheroidal Wave Functions GIP (PSWF-GIP) or Cross-Spectral Metric Smoothing GIP (CSMS-GIP), which also select appropriate training bins based on certain statistics and a pre-transformed clutter covariance matrix of the entire set of training bins [23,24].

It should be noted that these methods are highly effective for homogeneous clutter. This is due to the assumption that clutter bins have identical probability decompositions. However, statistical methods require access to a large number of training bins, which is mostly difficult to meet.

Currently, researchers are focusing all their efforts on improving non-static methods, which are computationally less complex methods in terms of both time and memory [25,26,27]. Nonetheless, statistical methods are subject to computational errors, which reduces the probability of accurate object detection.

When considering the development of non-statistical methods for determining the clutter covariance matrix, direct data domain STAP (D3-STAP) algorithms should also be mentioned. The idea behind this type of algorithm is to use a single cross-section of a radar data cube, that is, data collected for a single distance bin, to determine a vector of weights. Two typical D3 STAP algorithms are the direct data domain least square (D3 LS) algorithm and the direct data domain sparse recovery (D3 SR) algorithm, which uses the sparsity of the interference matrix [28,29,30,31].

Currently, on the basis of the latest literature, it is necessary to cite a completely new D3-STAP method proposed in the works [32,33], which is based on the knowledge of the Doppler frequency of the target and the speed of movement of the radar platform, as well as an optimization method that is used to determine the necessary distance bins required to reconstruct the clutter covariance matrix. The name of the proposed method is not given directly; however, for the purposes of this article, the acronym D3-STAP-ANM (D3-STAP based on atomic norm minimization) is adopted.

In the past few years, a whole series of studies on non-statistical methods for the estimation of the clutter covariance matrix has mainly focused on the combination of direct data domain (D3) STAP methods, together with sparse recovery algorithms, as evidenced by publications [34,35,36,37,38,39]. Thus, it has been shown that by using a small number of training bins or, in some cases, one, as well as by using sparsity of clutter in the spatial–temporal domain, it is possible to obtain a high resolution of the spatial–temporal clutter spectrum, which directly provides the estimation of the clutter covariance matrix.

To improve the target detection performance of airborne radar in the presence of clutter, adaptive real-time processing (STAP) with a single-input–multiple-output (SIMO) array (i.e., phased array) has been proposed [40,41,42]. The key of STAP is the use of an adaptive multivariate filter to create a deep indentation on the clutter ridge and thus effectively suppress the clutter. In recent years, it is noteworthy that the STAP technique has been increasingly used in MIMO radar systems [43,44,45,46,47]. Compared to conventional SIMO STAP, MIMO STAP can achieve better minimum detectable velocity (MDV) performance. This means that such a radar system is more capable of detecting slow-moving targets in the presence of strong clutter.

When analyzing research on MIMO, attention should be paid to [48], where a novel secrecy energy-efficient hybrid beamforming design for satellite–terrestrial integrated networks operating in the mmWave band was presented. In addition, article [49] examined satellite multicast communication and antenna-integrated networks with rate division multiple access, where satellite and unmanned aerial vehicle components are controlled by the network management center and operate in the same frequency band. In paper [50], the joint optimization design of beamforming and power allocation in the downlink non-orthogonal multiple access based for satellite–terrestrial integrated networks operating in the mmWave band was investigated. On the other hand, in article [51], a novel beamforming design for cognitive satellite–terrestrial networks operating in the mmWave band was presented.

As it turns out, in real-world conditions, determining the clutter covariance matrix and its inverse in this way is difficult. Furthermore, statistical algorithms fail when the data contained in the training bins do not reflect the statistical properties of the clutter of the bin under test, especially in an environment of heterogeneous clutter. These are natural drawbacks of our method, as well. Therefore, taking into account the extensive literature analysis presented in the introduction of our article, we noted that attention has now been focused worldwide on the study of non-statistical methods for estimating clutter covariance matrices in heterogeneous environments. Our goal was to break this convention and try to conduct research on new statistical methods for estimating clutter covariance matrices in STAP. It turns out that this possibility exists. Based on our knowledge of numerical methods, we saw the possibility of reducing the computational complexity and software implementation of the long-existing SMI method. This paper proves that it is possible to estimate the clutter covariance matrix using the MIMO radar geometry model and the statistical clutter matrix covariance method. The estimated clutter covariance matrix was determined using the LU method with partial pivoting. Hence, it can be said that our research fills a gap in current research on STAP.

The rest of this paper is organized as follows. Section 3 establishes the signal model and formulates the waveform design problem. Section 4 develops a new algorithm for STAP in MIMO radar in the presence of clutter. Section 5 provides numerical examples to demonstrate the performance of the proposed algorithm. Finally, Section 6 presents our conclusions.

## 3. Signal Model

The model of the MIMO radar mounted on a flying platform was taken into consideration. The MIMO radar consists of an *M*-element array of transmitting antennas and an *N*-element array of receiving antennas. Each element of a uniform linear array (ULA) is isotropic. The distances between the elements of the array of the transmitting antennas are equal to *d_t_*, and the distances between the elements of the receiving antenna array are equal to *d_r_*. The model assumes the transmission of orthogonal signals with time division multiplexing (TDM). The construction of the MIMO antenna array is shown in Figure 1.

Figure 2 shows the geometry of the radar system mounted on a flying platform moving along a straight line at a speed, *V_a_*. The distance from the radar to the object is *R_s_*. The radar observes the target at an angle, *α*. The radar system modeled in this way operates at a specific wavelength, λ, and transmits a sequence, *C*, of coherent pulses with a pulse repetition rate, *f_r_*. The presence of *P* targets was assumed, where the elevation angle between the radar and the *p*th object is *θ_p_* (*p* = 1, 2…, *P*).

There are *M*-matched filters in each of the *N*-receiving antennas to receive orthogonal signals from *M*-transmitting antennas. Hence, the total number of received signals is equal to the product of *MN*. The matrix of received data can be represented as [1,2,3,4]:(1)X=AS+Z 
(2)$$A=A_{r}⊙A_{t}\;$$
where denotes virtual array antennas, ***A_t_*** denotes the transmit steering matrix with the *p*th transmit steering vector, ***a_t_*** (*θ_p_*), and ***A_r_*** denotes the receive steering matrix with the *p*th receive steering vector, ***a_r_*** (*θ_p_*).
(3)$$A_{r}={\left[a_{r}\left(\theta_{1}\right),a_{r}\left(\theta_{2}\right),\ldots ,a_{r}\left(\theta_{p}\right)\;\right]}^{\;}\;$$
(4)$$a_{r}\left(\theta_{P}\right)={\left[1,e^{-j2\pi \frac{d_{r}\sin\theta_{P}}{\lambda }},\ldots ,e^{-j2\pi \left(N-1\right)\frac{d_{r}\sin\theta_{P}}{\lambda }}\right]}^{T}\;$$
(5)$$A_{t}={\left[a_{t}\left(\theta_{1}\right),a_{t}\left(\theta_{2}\right),\ldots ,a_{t}\left(\theta_{p}\right)\;\right]}^{\;}\;$$
(6)$$a_{t}\left(\theta_{P}\right)={\left[1,e^{-j2\pi \frac{d_{t}\sin\theta_{P}}{\lambda }},\ldots ,e^{-j2\pi \left(M-1\right)\frac{d_{t}\sin\theta_{P}}{\lambda }}\right]}^{T}\;$$

*S* denotes the data matrix from the received target; it also includes the clutter and jammer. ***X*** is the data matrix corresponding to a virtual array, with *x_m,n_* being the output data of the (*n* − 1) × *M* + *m*th virtual array element, where *m* = 1,2, …, *M* and *n* = 1,2, …, *N*. ***Z*** represents the white Gaussian noise matrix, with ***z_m,n_*** being the noise vector of the (*n* − 1) × *M* + *m*th virtual element.
(7)$$S={\left[s_{1},\;s_{2},\ldots ,s_{C}\right]}^{\;}\;$$
(8)$$X={\left[x_{1,1},\;\ldots ,x_{M,N}\right]}^{\;}$$
(9)$$x_{m,n}={\left[x_{m,n}\left(1\right),\;x_{m,n}\left(2\right),\ldots ,x_{m,n}\left(C\right)\right]}^{\;}$$
(10)$$Z=\left[z_{1,1},\;\ldots ,z_{M,N}\right]$$
(11)$$z_{m,n}={\left[z_{m,n}\left(1\right),\;z_{m,n}\left(2\right),\ldots ,z_{m,n}\left(C\right)\right]}^{\;}$$

The covariance matrix of the received data is given as [1]:(12)$$R_{x}=E\left\{XX^{H}\right\}$$
(13)$$R_{x}=AR_{s}A{\;}^{H}+R_{N}$$
(14)$$R_{s}=E\left\{SS^{H}\right\}$$
where ***R_s_*** denotes the covariance matrix of the received signal, and ***R_N_*** denotes the covariance matrix of the noise.

The STAP processor is a linear filter, so ultimately, the STAP processor is designed to remove noise and detect the target. The relationship describing these activities is given as [1,2,3,4]:(15)$$Y=w^{H}X$$
where ***w*** denotes the vector of weights given by the formula [1]:(16)$$w=R_{x}{}^{-1}A$$

In order to realize the essence of the issues discussed, it seems very helpful to present the STAP scheme. Hence, the following figure highlights the most important stages of the STAP radar signal processing (Figure 3).

## 4. New Method, SMI-LU

This section describes how to determine the clutter covariance matrix in the SMI method and a new method for determining the clutter covariance matrix based on the LU decomposition.

### 4.1. The Common Part of Both Algorithms

The previous section presented Formula (1), describing the echo of the received signal. Graphically, the received echo is represented by a radar data cube. To determine the vector of the weights given by Formula (16), it is necessary to find the clutter covariance matrix (12) using the SMI method. To do this, the SMI algorithm operates on the raw data by using a given cross-section of the radar cube at a fixed distance bin. Hypotheses about the presence or absence of a target in a given distance bin are then tested. For this purpose, a filter is created, characterized by high amplification of the useful signal from the target and, at the same time, high attenuation of all other signals (the clutter and jammer). More generally, STAP aims to filter out echoes coming from sources of interference and preserve the signal coming from the target of interest.

Figure 4 shows the estimation of the clutter covariance matrix using neighboring distance bins (the SMI method). In the figure, the test bin is marked, and guard bins are marked on either side of it. The covariance matrix used to calculate the optimal filter should not contain target data. For this reason, the distance bin data, in which the target location is expected, is not used. Therefore, guard bins are used to prevent the use of the data contained in the distance bin under test. The set of distance bins around the target test bin is called training bins. Finally, after selecting the correct training bins, the covariance matrix is determined. The clutter covariance matrix is then inverted.

The following is an algorithm for determining the set of training bins based on the known number of distance bins and the number of guard bins, the required number of training bins, and the known number of the distance bin under test. For this purpose, the following designations were adopted:*K*—number of distance bins;*K_T_*—number of training distance bins;*K_O_*—number of guard bins;*K_TARGET_*—distance bin under test;*I_KOP_*—lower index of guard bins;*I_KOZ_*—upper index of guard bins;*K_BEFORE_*—the remaining number of training bins before *K_TARGET_*;*K_AFTER_*—the remaining number of training bins behind *K_TARGET_*;*K_Tbefore_*—number of training bins before *K_TARGET_*;*K_Tafter_*—number of training bins behind *K_TARGET_*;*I_Kbefore_*—distance training bin indexes before *K_TARGET_*;*I_Kafter_*—distance training bin indexes behind *K_TARGET_*;

Let the initial value [1,2,3] be:(17)$$I_{KOP}=\;K_{TARGET}-\frac{K_{O}}{2}$$
(18)$$I_{KOZ}=\;K_{TARGET}+\frac{K_{O}}{2}$$
(19)$$K_{BEFORE}=\max\left(I_{KOP}-1,\;0\right)$$
(20)$$K_{AFTER}=\max\left(K-I_{KOZ},\;0\right)$$

If $K_{BEFORE}<K_{AFTER}$, let the initial value:(21)$$K_{Tbefore}=\;\min\left(\frac{K_{T}}{2},K_{BEFORE}\;\right)$$
(22)$$K_{Tafter}=\;\min\left(K_{T}-K_{Tbefore},K_{AFTER}\;\right)$$

If $K_{BEFORE}>K_{AFTER}$, let the initial value:(23)$$K_{Tafter}=\min\left(\frac{K_{T}}{2},K_{AFTER}\;\right)$$
(24)$$K_{Tbefore}=\;\min\left(K_{T}-K_{Tafter},K_{BEFORE}\;\right)$$

Finally, the training data matrix should be completed considering the data from the radar data cube with range bins indexes equal to
(25)$$I_{Kbefore}=I_{KOP}+\left(-K_{Tbefore}:-1\;\right)$$
(26)$$I_{Kafter}=I_{KOZ}+\left(1:K_{Tafter}\right)$$

### 4.2. SMI-QR

Based on the ***D_T_*** training data matrix selected according to the above algorithm, the essential part of the standard SMI algorithm follows, namely the determination of the vector of the weights, w.

To do this, the first step is to decompose the QR training data matrix, DT, into an orthogonal matrix, ***Q***, an upper triangular matrix, ***R***, and a vector, ***P***, such that [4]:(27)$$D_{T}\cdot P=Q\cdot R\;$$

Then, determine the matrix, ***F***, according to the relationship:(28)$$F=\frac{R^{T}}{A^{*}}\;$$
whereby $A^{*}$ means that only as many rows of matrix ***A*** are taken into account as is equal to the number of columns of matrix ***P***. Finally, the clutter covariance matrix is determined:(29)$$R_{x}=\frac{R}{F}$$

The clutter covariance matrix thus determined is then used to determine the vector of the weights, according to relation (16).

### 4.3. SMI-LU

Based on the *D_T_* training data matrix selected according to the SMI algorithm, the essential part of the proposed SMI-LU algorithm follows; that is, the determination of the weight vector, w, using LU decomposition with partial pivoting. LU decomposition with partial pivoting is numerically stable and was chosen because LU decomposition methods fail when a zero element is encountered by which to perform the division. In such cases, the algorithms abort the calculation. Meanwhile, the solution exists and is quite simple. In such a case, it is enough to rearrange the rows of the decomposed matrix so that the elements with the largest modules are located on its diagonal.

To do this, the first step is to decompose the LU of the ***D_T_*** training data matrix into a lower triangular matrix, ***L***, and an upper triangular matrix, ***U***. Then, determine the clutter covariance matrix:(30)$$R_{x}=\frac{U}{L^{T}\cdot A}\;$$

The clutter covariance matrix thus determined is then used to determine the vector of the weights, according to relation (16).

## 5. Simulation

This paragraph presents a series of simulation studies in the MATLAB environment that, according to the authors, provide a significant basis for evaluating the validity of the proposed SMI-LU solution. Table 1 contains all the values of the parameters adopted for the simulation. In order to compare the performance of the proposed algorithm, the simulation results are compared under four algorithms based on determining the clutter covariance matrix from among the data contained in the radar data cube, i.e., the SMI algorithm, displaced phase center array (DPCA) algorithm, adaptive displaced phase center array (ADPCA) algorithm, and the proposed SMI-LU algorithm. A description of the DPCA and ADPCA algorithms can be found in [52,53].

### 5.1. Improvement Factor

The basic parameter that determines the performance of any STAP linear processor in the literature is called the improvement factor (IF), which defines the quotient of the signal-to-noise (SNR) power ratio at the output and input of the optimal processor. Taking into account the cited definition of the IF parameter, it is worth simulating the effect of the number of pulses on the value of the IF of the proposed method.

Accordingly, the following figure shows the effect of the number of pulses processed by the STAP processor on the performance of the processor as measured by the IF coefficient. The analysis of the obtained results clearly shows that the more pulses transmitted by the radar need to be processed in the STAP processor, the higher the performance of the processor.

Figure 5 shows the simulation for the case when the receiver’s own noise-to-interference ratio was −20 dB. If one assumes for the simulation a receiver noise-to-interference ratio of −30 dB (Figure 6), which means less influence of the receiver’s own noise, it turns out that the performance of the STAP processor improves significantly. From an engineering point of view, it can be concluded that it is worthwhile to use low-noise components in the radar’s receiving path, which will significantly improve the performance of the STAP processor and thus have a positive effect on target detection.

It was necessary to check the performance of the proposed SMI-LU method in terms of clutter suppression performance based on the IF. Figure 7 shows the clutter suppression performance for the four STAP algorithms used.

The proposed method uses LU decomposition, for which better clutter suppression can be achieved compared to the same STAP, but by using QR decomposition (SMI) or other methods, the IF curve indentation is narrower and reaches higher values. Furthermore, the implementation of the SMI algorithm based on QR decomposition is much more complicated programmatically, which also affects its practical application. The STAP-LU algorithm is much simpler computationally, which is a direct result of the previously cited steps of both algorithms, which consist of numerous operations performed on the matrices.

The IF of more than 70 dB is a very good result in the context of the IF parameter results boasted by researchers in recent publications [21,22,23,24]. Moreover, it provides evidence of proper clutter cancellation. It can be concluded that the developed algorithm and its software implementation in the Matlab environment encourage one to go a step further and try practical verification of the proposed solution. Unfortunately, this involves the outlay of considerable financial costs.

### 5.2. Signal-to-Interference Plus Noise Ratio

Another significant parameter characterizing the performance of any linear STAP processor is the output signal-to-interference plus noise ratio (SINR) of the processor. Figure 8 shows a plot of the SINR at the output of the STAP processor for all four methods. Considering the SMI-LU algorithm, the processor achieves an SINR value of about 35 dB over almost the entire Doppler frequency range. If the Doppler frequency of the object is close to 0 Hz or 30 kHz, then the SINR value is very small, because at these frequencies, the algorithm very strongly eliminates interference; hence, the signal-to-interference plus noise ratio is also very low.

### 5.3. Precise Target Detection

In order to validate the correctness of the proposed SMI-LU method using LU decomposition and the MIMO radar system, another simulation was conducted. The key assumption made in this simulation was the presence of an environment characterized by inhomogeneous interference, so the frequently used model of inhomogeneous forest-covered terrain, referred to in the MATLAB environment as the gamma model, was adopted as the clutter.

Figure 9 shows the values of the received signals by the MIMO radar antenna array as a function of the range after transmitting the first pulse. At this stage, the received signals form a data cube, which has not yet been processed by the newly developed STAP algorithm. As a result, the radar against a background of strong clutter is unable to indicate the location of a target. As can easily be seen, the radar erroneously indicates that the target is at a distance of 1000 m from the radar.

Figure 10 shows the values of the received signals by a MIMO radar antenna array as a function of range after transmitting the first pulse. However, this time, the raw data were subjected to STAP by implementing the proposed SMI-LU algorithm in the MATLAB environment. As can easily be seen, the radar correctly indicates that the target is at a distance of about 1900 m from the radar in a straight line.

The execution of the latest simulation was guided by two goals. The first goal was to fully implement the proposed SMI-LU algorithm and environment along with a flying platform with radar designed to detect the target, as well as eliminate non-uniform clutter. The second goal was to recreate as real an environment as close to the real one as possible. This was due to the lack of research conducted practically. The obtained precise target data lead us to believe that the proposed STAP algorithm can be successfully applied in practical implementations.

## 6. Conclusions

In this paper, the theoretical basis of STAP was presented first. The individual steps of the classical SMI STAP algorithm were discussed. The importance of the characteristic parameters describing the STAP linear processor was highlighted by discussing the performance parameters, and the most important features of MIMO radar were presented.

This was followed by a critical analysis of the past and current methods of estimating the clutter covariance matrix. Statistical and non-statistical methods were compared. An area of research development on current methods was indicated, which prompted the search for another method of estimating the clutter covariance matrix. As a result of a series of analyses, the third chapter presents a description and course of a novel method for estimating the clutter covariance matrix based on LU decomposition.

Current analyses of the literature each unequivocally state that various methods of estimating the clutter covariance matrix are appropriate in STAP, with a move away from statistical methods. Currently, the most developed are non-statistical methods for estimating the clutter covariance matrix. However, the authors of the article noted the possibility of reducing the computational complexity and software implementation of the SMI method, which has existed for years.

This article is devoted to the presentation of a new statistical STAP algorithm based on LU decomposition. The theoretical analyses performed in the course of the work, the calculations, and the obtained results of simulation studies allowed us to make a clear conclusion that the SMI-LU method effectively eliminates clutter in heterogeneous environments, which should be considered a great advantage.

In conclusion, this paper proves that it is possible to estimate the covariance matrix of STAP clutter by using the MIMO radar geometry model and SMI-LU algorithm.

## Figures and Tables

**Figure 1 sensors-23-04280-f001:**
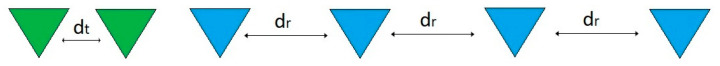
Example of MIMO array obtained via M = 2 and N = 4 antennas.

**Figure 2 sensors-23-04280-f002:**
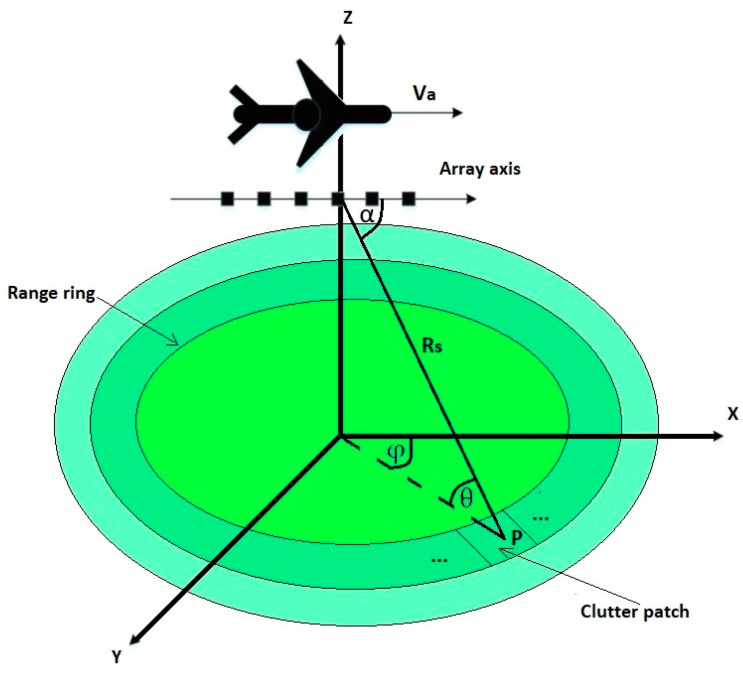
Geometry of MIMO radar system.

**Figure 3 sensors-23-04280-f003:**
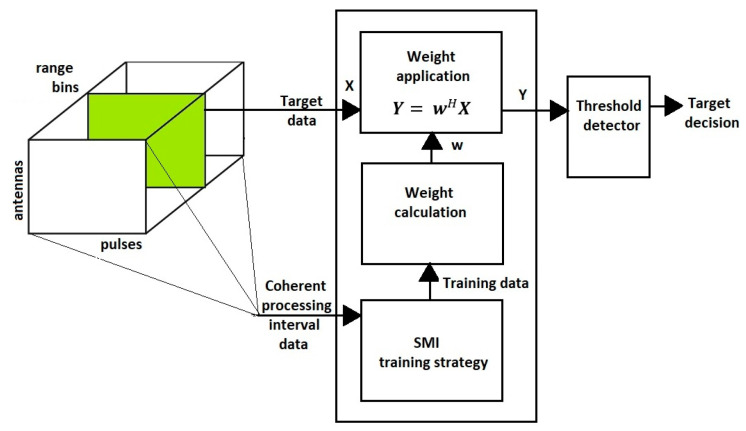
STAP.

**Figure 4 sensors-23-04280-f004:**
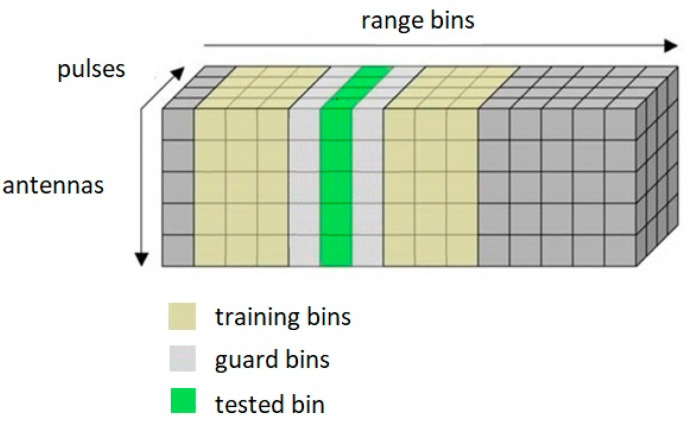
Radar data cube—SMI training strategy.

**Figure 5 sensors-23-04280-f005:**
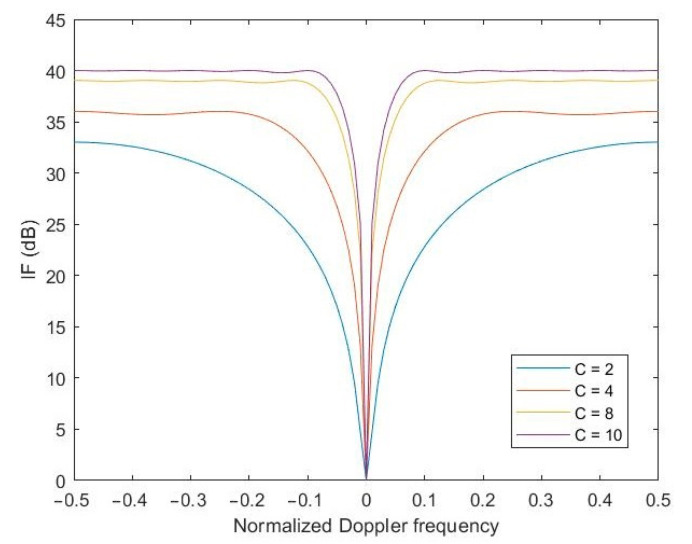
Diagram of the IF depending on the number of pulses. The ratio of the receiver’s own noise-to-interference equal to −20 dB.

**Figure 6 sensors-23-04280-f006:**
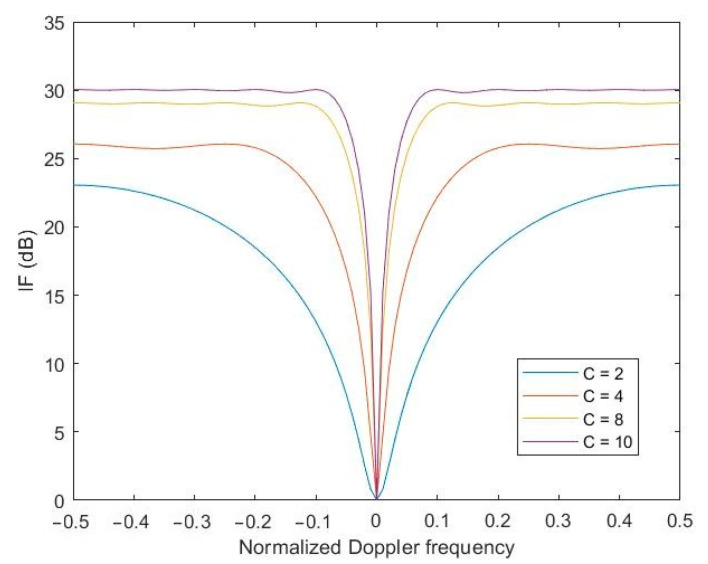
Diagram of the IF depending on the number of pulses. The ratio of the receiver’s own noise-to-interference equal to −30 dB.

**Figure 7 sensors-23-04280-f007:**
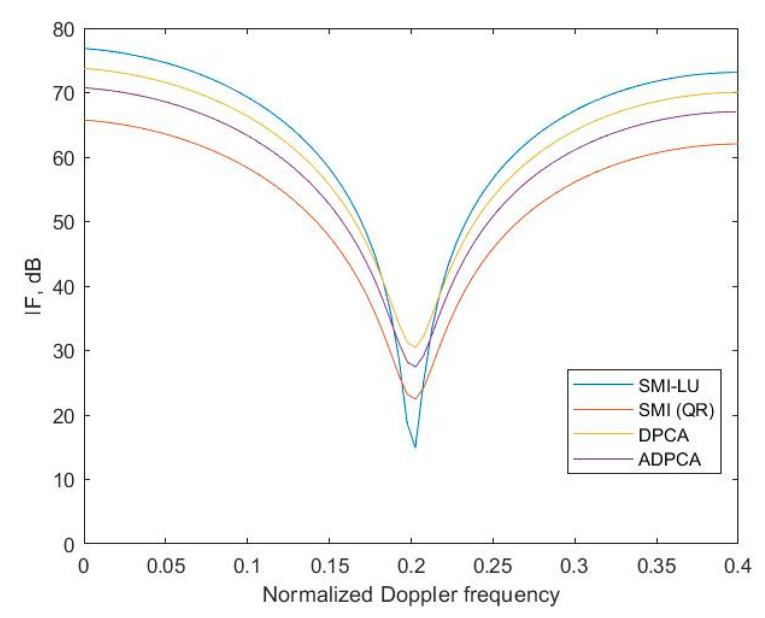
Performance comparison of STAP algorithms.

**Figure 8 sensors-23-04280-f008:**
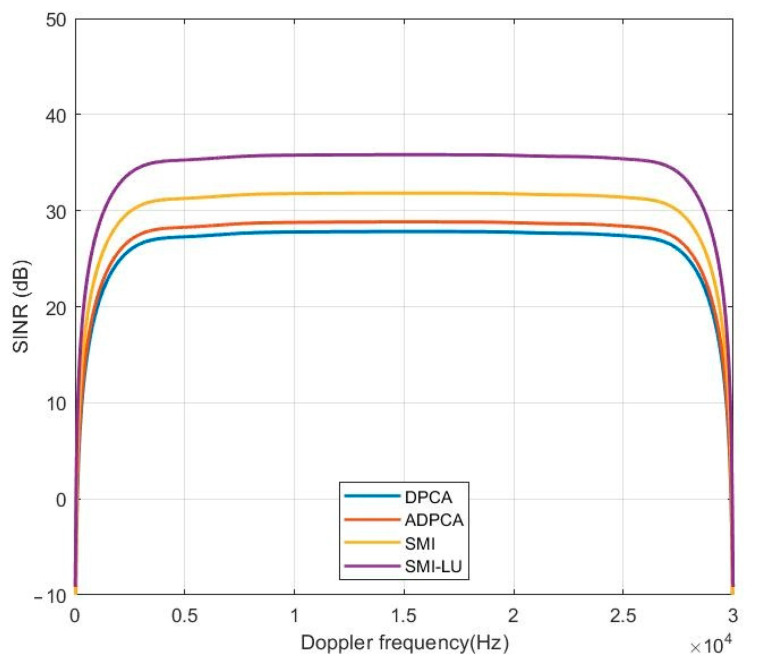
Plot of SINR at the output of the STAP processor.

**Figure 9 sensors-23-04280-f009:**
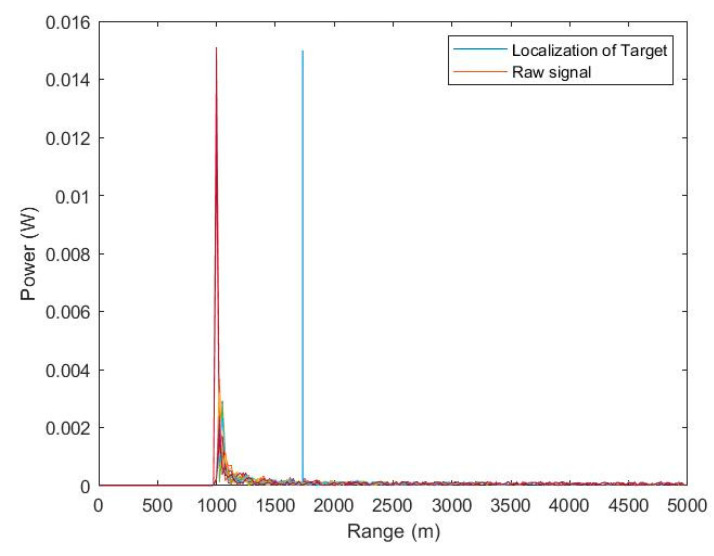
Values of received signals by the MIMO radar antenna array as a function of range after transmitting the first pulse before STAP.

**Figure 10 sensors-23-04280-f010:**
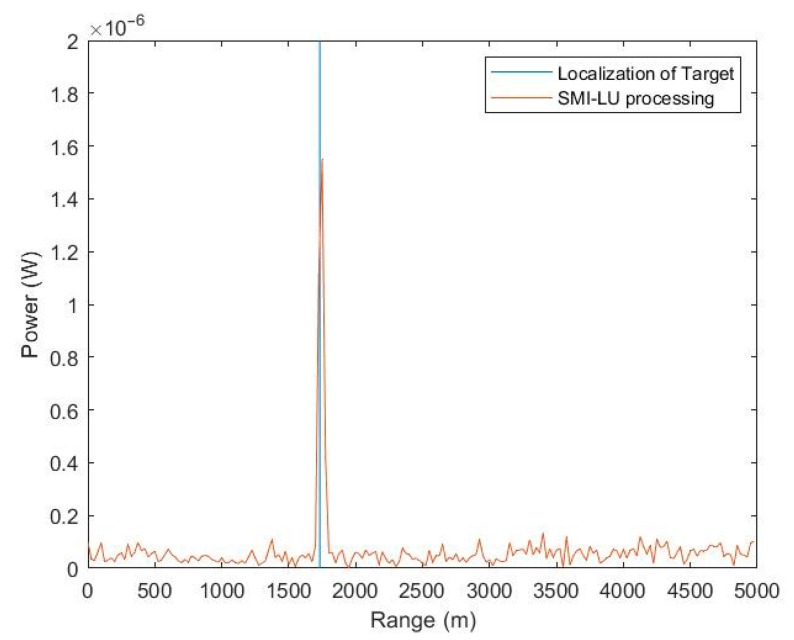
Values of received signals by the MIMO radar antenna array as a function of range after transmitting the first pulse and SMI-LU processing.

**Table 1 sensors-23-04280-t001:** Parameters adopted for simulation.

Parameter	Value
Radar operating frequency	10 GHz
Number of transmitting antennas	5
Number of receiving antennas	5
Pulse repetition frequency	30 kHz
Pulse duration	33 µs
Number of pulses	10
Range	5000 m
Transmitter power	5.6 kW
Directional gain	20 dB
Platform height	1000 m
Speed of movement of the platform	[160 m/s, 160 m/s, 0]
Speed of movement of the target	[30 m/s, 30 m/s, 0]
Initial localization of the platform	[0,0,1000]
Initial localization of the target	[0,1000,1000]

## Data Availability

All Data are available upon reasonable request made to the corresponding author.

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
