# Peer review of "A New Statistical Method for Determining the Clutter Covariance Matrix in Spatial–Temporal Adaptive Processing of a Radar Signal"

_sensors, 2023, doi:10.3390/s23094280_

Round 1
Reviewer 1 Report
The paper proposes a statistical method for determining the clutter covariance matrix in spatial-temporal adaptive processing of a radar signal using LU decomposition. Analyzing the paper, I identified the following issues:
1. Introduction section is very long, so I suggest to be split in two sections: Section 1. Introduction and Section 2. Related Work;
2. Subsection 3.3 needs to be enriched with more details. For example, the simplest variant of LU is numerically unstable in practice, and needs to be enhanced (see for example LU factorization with partial pivoting) to be numerically stable;
3. Line 134: it is written “However, the paper's authors noted” is confusing since the reader cannot understand to what paper it refers. I suggest “However, we noted”;
4. Lines 139-143: instead of Arabic numerals (e.g. Section 2, Section 3), Roman numerals are used (e.g. Section II Section III).
5. Line 240: it is written “distribution” instead “decomposition”;
6. The name of the columns in Table 1 are not in English.
7. Last line in Table 1: I suggest to delete definite article “the” since the other rows in Table 1 do not include it;
8. In the simulation part, it will be beneficial to compare the variant that uses QR with the one that uses LU.
9. Line 138: why the authors say that the LU method is "modified"? It seems that they didn't modify this method.
English is generally fine. It only needs some polishing (see typos in Table 1).
Reviewer 2 Report
Please see the attached comments.

Moderate editing of English language needs to be checked.
Reviewer 3 Report
The authors present a novel statistical method for determining the clutter covariance matrix. The paper is quite well organised and the results have some novelty.
I recommend it for publication after a minor revision.
Discuss the present research gaps and the limitations of the proposed statistical method.
Can we obtain a matrix or linear operator representation of the MIMO radar?
Discuss the advantages of LU decomposition.
Minor improvements needed.
Round 2
Reviewer 1 Report
The authors have successfully solved all my comments and concerns.
Reviewer 2 Report
The authors have well addressed all my concerns, no further comments.